# Tobramycin Nanoantibiotics and Their Advantages: A Minireview

**DOI:** 10.3390/ijms232214080

**Published:** 2022-11-15

**Authors:** Mariella Rosalia, Enrica Chiesa, Erika Maria Tottoli, Rossella Dorati, Ida Genta, Bice Conti, Silvia Pisani

**Affiliations:** 1Department of Drug Sciences, University of Pavia, Viale Taramelli 12, 27100 Pavia, Italy; 2Otorhinolaryngology Unit, Department of Surgical Sciences, Fondazione IRCCS Policlinico San Matteo, 27100 Pavia, Italy

**Keywords:** Tobramycin, multidrug resistance, nanosized carrier, liposomes, niosomes, polymer nanoparticles, hydrogels, nanofibers, nanomedicine

## Abstract

Nowadays, antimicrobial resistance (AMR) represents a challenge for antibiotic therapy, mostly involving Gram-negative bacteria. Among the strategies activated to overcome AMR, the repurposing of already available antimicrobial molecules by encapsulating them in drug delivery systems, such as nanoparticles (NPs) and also engineered NPs, seems to be promising. Tobramycin is a powerful and effective aminoglycoside, approved for complicated infections and reinfections and indicated mainly against Gram-negative bacteria, such as *Pseudomonas aeruginosa*, *Escherichia coli*, *Proteus*, *Klebsiella*, *Enterobacter*, *Serratia*, *Providencia*, and *Citrobacter* species. However, the drug presents several side effects, mostly due to dose frequency, and for this reason, it is a good candidate for nanomedicine formulation. This review paper is focused on what has been conducted in the last 20 years for the development of Tobramycin nanosized delivery systems (nanoantibiotics), with critical discussion and comparison. Tobramycin was selected as the antimicrobial drug because it is a wide-spectrum antibiotic that is effective against both Gram-positive and Gram-negative aerobic bacteria, and it is characterized by a fast bactericidal effect, even against multidrug-resistant microorganisms (MDR).

## 1. Introduction

It is well known worldwide that antibiotic therapy was the cornerstone to the reduction in morbidity and mortality due to bacterial infections in the last century. However, today, important changes in society, technological progress, and the evolution of microorganisms themselves are contributing to an increase in emerging and re-emerging infectious diseases and to the appearance of antimicrobial resistance (AMR) and even multidrug resistance (MDR). According to the European Centre for Diseases Prevention and Control (Ecdc) report, dated January 2022, more than 670,000 drug-resistant bacterial infections occurred in the European Union (EU) and European Economic Area (EEA) alone, and approximately 33,000 people died as a direct consequence of these infections, the health burden of AMR is comparable to that of influenza, tuberculosis, and HIV/AIDS combined [1]. The cases of infection caused by antibiotic-resistant bacteria are steadily increasing and are expected to kill 10 million people per year by 2050 [2]. It can be stated that AMR has reached critical levels, continuing to invalidate the action of the majority of antibiotic drugs currently used in the clinical field.

Antimicrobial resistance is a natural phenomenon by which microorganisms adapt to survive environmental threats, leading to their capability of surviving even in the presence of concentrations of antimicrobials normally inhibiting or killing microorganisms of the same species [3]. AMR is recognized to be a global health crisis that poses a great threat to modern medicine. It involves either Gram-negative or Gram-positive bacteria, planktonic, and biofilm infections. The mechanisms for AMR generation and spreading are well-known by scientists and extensively reported in the literature. F. Bernini and coll. summarize them well as (i) the modification and inactivation of the antimicrobial molecules by enzymes produced by the bacteria (i.e., bacteria resistant to β-lactams thanks to β-lactamase production); (ii) alterations of the cellular antibiotic target leading to binding prevention (i.e., by synthesis of a modified cell wall resistant to glycopeptide antibiotics); (iii) the expression of multidrug efflux pumps against many different classes of antibiotics (i.e., tigecycline or imipenem resistance in *Acinetobacter baumannii*); (iv) the reduction in cell permeability to preclude antibiotics reaching their target (particularly in Gram-negatives, such as *A. baumannii* and *Pseudomonas aeruginosa* due to the down-regulation or elimination of transmembrane porins) [4]. The main causes of AMR are the misuse and/or the overuse of antimicrobials that promote the transmission via horizontal transfer of resistance-conferring genes. This has been reported either for human pathologies, i.e., antibiotic consumption increased by 65% between 2000 and 2015 in both low- and middle-income countries, or for the misuse of antibiotics for animal growth promotion in farms and aquaculture [5,6].

Effective prevention strategies are urgently required to slow the emergence and further dissemination of AMR. The different strategies activated to overcome AMR can be summarized as the discovery of new antimicrobial molecules, such as antibacterial antibodies and antimicrobial peptides; the repurposing of already available antimicrobial molecules by encapsulating them in drug delivery systems such as nanoparticles (NPs) and also engineered NPs; the development of vaccines, faecal microbiota transplantation, and combined therapies [7,8,9,10,11]. Moreover, given the availability of data sets encompassing hundreds or thousands of pathogen genomes, machine learning (ML) is also increasing as a strategy to predict resistance to different antibiotics in pathogens based on their gene content and genome composition [12].

As long as the first two strategies are concerned, due to regulatory constraints, only 17 new systemic antibiotics have been approved in the US in the time span 2010–2021 [13], and several authors agree that the repurposing of already in use antimicrobial drugs through nano drug delivery systems is a good strategy to improve antimicrobial and anti-biofilm activities, also overcoming AMR [4,14,15].

Nanomedicines are defined by the European Medicine Agency as: “purposely designed systems for clinical applications, with at least one component at the nanoscale, resulting in reproducible properties and characteristics, related to the specific nanotechnology application and characteristics for the intended use (route of administration, dose), associated with the expected clinical advantages of nano-engineering (e.g., preferential organ/tissue distribution) and needs to meet the definition of a medicinal product according to European legislation”. In these terms, the nanoscale scale ranges from the atomic level at around 0.2 nm (2 Å) up to around 100 nm. Therefore, drug delivery systems based on nanoparticles are supposed to have peculiar properties derived from nanoparticle size and/or composition and architecture. Several examples of nanomedicines are already on the market and in the study, but the amazing one is represented by COVID-19 vaccines. It should be noted that mRNA sequence and modification were selected by designing mRNA with modified nucleotide sequences and mRNA capping modalities in order to stabilize it and its translation capacity. However, only through the vehiculation of the modified mRNA into lipid nanoparticles (LNPs), whose composition was attentively studied in order to formulate a stable mRNA nanomedicine, it was possible to achieve an effective and safe vaccine. Moreover, the LNPs exploit an immunogenic effect themselves and enhance mRNA delivery to the cytosol of antigen-presenting cells [16,17].

As long as antibiotics are concerned, nanoparticles as antibiotic drug delivery systems have been advantageous in overcoming AMR and also improving antibiotic activity towards bacteria biofilm, and several reports in the literature are available on this topic, typically referring to nanoantibiotics [4,15,18]. The nano drug delivery systems for antibiotic therapies encompass polymer nanoparticles, either made from natural (chitosan and derivatives) or synthetic (polylactide-co-glycolide) polymers, solid lipid nanoparticles, liposomes, and niosomes. The last two seem to be preferred because of their similarity with biological membranes, which some authors have indicated to be a preferential characteristic for cell uptake. The antibiotics investigated and loaded into nanoparticulate drug delivery systems are either drugs acting against Gram-positive or Gram-negative bacteria, such as glycopeptide antibiotics (vancomycin, teicoplanin), ciprofloxacin, colistin, and erythromycin [19,20,21,22].

Starting from this background, this review paper is focused on what has been conducted in the last 20 years for the development of Tobramycin nanosized delivery systems (nanoantibiotics) with a critical discussion and comparison. Tobramycin was selected as the antimicrobial drug because it is a wide-spectrum antibiotic that is effective against both Gram-positive and Gram-negative aerobic bacteria, and it is characterized by a fast bactericidal effect, even against multidrug-resistant microorganisms [23].

## 2. Tobramycin

Tobramycin is an aminoglycoside antibiotic (Figure 1) derived from *Streptomyces tenebrarius* with a bacteriostatic activity toward aerobic Gram-negative bacteria. The molecule is freely soluble in water, and its molecular weight is 467.5 g/mol. Following its active transport into bacterial cells, tobramycin binds irreversibly to 16S ribosomal RNA of the bacterial 30S ribosomal unit and interferes with the initiation complex between the messenger RNA and the 30S subunit, thereby inhibiting the initiation of protein synthesis, consequently leading to bacterial cell death. In addition, tobramycin induces the misreading of the mRNA template causing incorrect amino acids to be incorporated into the growing polypeptide chain, consequently interfering with the protein elongation. It is co-administered with beta-lactams that it penetrates the cell walls of aerobic Gram-negative bacteria and obtains a synergistical bactericidal effect. Tobramycin is a powerful and effective aminoglycoside, approved for complicated infections and reinfections, and is indicated mainly against Gram-negative bacteria, such as *Pseudomonas aeruginosa*, *Escherichia coli*, *Proteus*, *Klebsiella*, *Enterobacter*, *Serratia*, *Providencia*, *Citrobacter* species, and against certain Gram positives, including *Staphylococcus aureus* (penicillinase and no penicillinase-producing strains). Tobramycin treatable infections could range from septicaemia, lower respiratory tract infections, central nervous system (CNS) infections, such as meningitis, intra-abdominal infections, skin, and subcutaneous tissue infections, including osteomyelitis, and complicated urinary tract infections. Moreover, Tobramycin is quite commonly used to treat ocular infections through the topical administration of eye drops. Inhaled tobramycin is FDA approved to manage *P. aeruginosa* infections in cystic fibrosis patients from up to six years of age or older.

Tobramycin is more frequently administered via intravenous or intramuscular injection. Other administration routes are applied, such as ocular administration, but oral administration is avoided because the drug has poor oral absorption. Similar to other aminoglycosides, Tobramycin absorption is prevented by the efflux P-glycoprotein pump located in the brush border of the intestinal epithelial cells. The combination with beta-lactams, such as penicillin or cephalosporins, promotes Tobramycin penetration through the outer walls of Gram-negative bacteria, enhancing the antibacterial efficiency.

Tobramycin is absorbed rapidly following intramuscular injection, with peak serum concentrations achieved between 30 and 90 min. Therapeutic levels of tobramycin are generally considered between 4 and 6 mcg/mL after the administration of a 1 mg/kg of body weight dose. Tobramycin injected by intravenous infusion over one-hour results in similar serum concentrations to the intramuscular administration. The drug is excreted via the kidneys through glomerular filtration, and its serum half-life ranges from about 2 h in adults to about 4.5 to 8.7 h in neonates [24].

The most common Tobramycin side effects are dizziness, headache, confusion, nausea, and skin rash. Because of its weak binding affinity to the rRNA of eukaryotes, Tobramycin can cause serious adverse events in patients, including ototoxicity, neuropathy, and nephrotoxicity. Thus, the concurrent use of other nephrotoxic and neurotoxic antibiotics, especially other aminoglycosides (e.g., streptomycin, amikacin, kanamycin, neomycin, paromomycin, and gentamicin), cephaloridine, polymyxin B, viomycin, colistin, vancomycin, and cisplatin should be avoided, as well as potent diuretics, such as furosemide and ethacrynic acid. Longer dosing intervals may reduce the risk of ototoxicity and nephrotoxicity. Therefore, monitoring drug frequency and dosing are essential in tobramycin administration. The peculiar characteristics of Tobramycin make it an interesting candidate for the modified release of drug delivery systems to reduce its toxicity and to better control dosing. In these terms, nanosized drug delivery systems, such as liposomes, have been studied since they might improve antibiotic activity.

## 3. Nanosized Carriers for Tobramycin

Figure 2 shows various examples of nanosized delivery systems that have been studied and reviewed for antibiotics, such as polymer and functionalized polymer nanoparticles, polymer-drug conjugated, dendrimers, polymer hydrogels, niosomes, liposomes, solid lipid nanoparticles, phospholipid micelles, and inorganic nanoparticles [14]. The first studies on aminoglycosidic antibiotics delivery through nanosized drug delivery systems date from the late 1990’s [25]. As long as Tobramycin is concerned, a wide range of reports in the literature is available on its encapsulation in nanosized drug delivery systems. According to a search on the Web of Science (Figure 3A), a total of 171 papers have been published dealing with Tobramycin nanoparticles since the year 2000.

The search metrics highlight that 86 out of 171 publications deal with Tobramycin-loaded liposomes, 26 are related to polymer nanoparticles and 11 to solid lipid nanoparticles loaded with Tobramycin (SLN). Therefore, it can be stated that liposomes are the preferential drug delivery system for Tobramycin. Moreover, a dramatic increase in publications on Tobramycin nanoparticles is highlighted starting from the year 2018. 

More recently, Tobramycin encapsulation in nanofiber and nanofibrous scaffolds for tissue regeneration has been proposed to achieve delivery systems for the localized administration of Tobramycin, even as a support for preventing infections in the scaffolds after implantation. A Web of Sciences search highlighted five publications on this topic starting from 2015, showing that this is a more specialized delivery of antibiotics with increasing interest by the scientific community (Figure 3B). 

Table 1 reports non-exhaustive examples from the literature search on Tobramycin-loaded nanocarriers, subdivided by the type of carrier and highlighting their composition and antibacterial activity, which is mainly addressed against Gram-negative microorganisms, namely *P. aeruginosa*. These studies are consistent in reporting Tobramycin as an antibiotic of choice in the treatment of cystic fibrosis (CF) associated *P. aeruginosa* infections and even by localized administration of Tobramycin, as delineated in the literature [26,27]. Tobramycin inhalation solution and tobramycin inhalation powder are commercially available for the treatment of chronic lung infections caused by *P. aeruginosa*. However, the efficacy of the free drug administration in CF patients is not sufficient to achieve therapeutic levels at the site of the infection due to its rapid clearance and poor mucus penetration. Most efforts in the literature have been paid to improve the efficacy of localized administered Tobramycin through its nanoencapsulation, also promoting drug penetration into the bacterial biofilm and mucus. In the following sections, Tobramycin encapsulation in nanosized delivery systems and fibres is described and discussed.

### 3.1. Liposomes

Tobramycin was formulated and delivered mainly through the liposomes and niosomes with the aim of increasing its antimicrobial activity against Gram-negative bacteria and biofilms [28,29,31,32,33,38,39]. Moreover, a drug product based on liposomal Tobramycin for inhalation (Axentis Pharma AG) was designed as an orphan drug in 2021 and was phase II experimented.

Marier and coll. evaluated a liposome formulation for the localized administration of Tobramycin against pulmonary infections of *Burkholderia cepacia* and *Pseudomonas aeruginosa*. Localized administration has the advantage of delivering the drug at the site of infection, reducing unnecessary systemic exposure and, consequently, side effects. The liposomes’ formulation included two synthetic phospholipids, namely noncharged dipalmitoylphosphatidylcholine phospholipid and negatively charged dimyristoylphosphatidylglycerol phospholipid, used in a 10:1 molar ratio. The liposomes were manufactured by thin film hydration methods and were administered intratracheally to rats. In both cases, the liposomal formulation of Tobramycin promoted drug retention in the lungs and improved its efficacy after multiple treatments [28,29]. These results supported the use of liposomal Tobramycin to treat either *Pseudomonas aeruginosa* pulmonary infections in cystic fibrosis patients or *Burkholderia cepacia infection*. The encapsulation of the drug changed its pulmonary pharmacokinetic profile, resulting in a slower distribution and elimination. Moreover, Tobramycin-loaded liposomes demonstrated an ability to penetrate the bacterial biofilm. Bacterial biofilm production is typical in cystic fibrosis, mainly originating from *P. aeruginosa* itself. Generally speaking, bacterial biofilms are clusters of microorganisms adhering to a surface and/or to each other and are embedded in a self-produced extracellular matrix. In cystic fibrosis, the biofilm matrix consists of substances, such as proteins (e.g., fibrin), polysaccharides (e.g., alginate), as well as eDNA from mucoid bacteria, mucins from lung epithelial cells, and DNA from damaged leucocytes. The matrix production is promoted by the hypoxic environment: the embedded bacteria are affected by the lack of oxygen and bacterial gene expression modifications, and metabolism and protein production alterations are induced, leading to a lower metabolic rate and a reduced rate of cell division. As a consequence, the biofilm viscous matrix develops, and a physical barrier to drug penetration is obtained so that the modified bacteria are more resistant to antibiotic drugs. Bacterial biofilm formation is typical of chronic infections, such as cystic fibrosis, where there is stationary mucus. Biofilms are very difficult to be eradicated and can ultimately lead to tissue damage and acute infection [14,40,41]. Liposomes have been widely investigated for their potential to invade and control infectious biofilms due to their unique characteristics, such as cell membrane-fusogenicity, biodegradability, versatility, biocompatibility, and low toxicity [42,43]. Typical features of liposomes that promote their penetration into bacterial biofilms are their size, around 150–300 nm (which is lower than the bacterial dimensions), and their stability, preferably with a cationic surface charge of around +30 mV. More recently, Tobramycin was formulated in the liposomes when combined with other substances, such as N-acetylcysteine or bismuth-ethanedithiol, with the aim to improve its antibiofilm activity [30,31,32,33]. Reem E. Alarfaj and coll. investigated the co-encapsulation of Tobramycin and N-acetylcysteine into the liposomes (TNL) made from 1,2-Dimyristoyl-sn-glycero-3-phosphoethanolamine (DMPE), 1,2-Dipalmitoyl-sn-glycero3-phosphocholine (DPPC), and cholesterol with the aim to evaluate if the N-acetylcysteine mucolytic effect was synergistic to Tobramycin antibiotic effects. The liposomes were around 250–300 nm, and TNL had a lower size with respect to the liposomes loaded with Tobramycin alone (TL), while the encapsulation efficiency increased. Moreover, TNL was more stable than TL. The authors tested TL and TNL activity against clinical strains of resistant *E. coli*, *K. pneumoniae*, and *A. baumannii* in the presence of several resistance genes. They found that both TL and TNL formulations reduced the MIC and MBC against most of the clinical isolates, but TNL had a higher efficacy, demonstrating the synergistic effect of N-acetylcysteine [33].

Nouf M. Alzahrani and coll. investigated the antibacterial and antibiofilm activity of liposomes loaded with Tobramycin and co-encapsulated with anti-biofilm peptide (IDR 1018) against *P. aeruginosa* on multidrug-resistant isolates (MDR 7067). Tobramycin was selected among three different antibiotic drugs because it resulted in the one with the highest antimicrobial activity. The peptide is a cationic synthetic derivative of bactenecin, a bovine host-defence peptide (HDP), which could facilitate the disruption of bacterial biofilms and thus increase bacterial killing. The authors prepared cationic liposomes from Cationic lipid 3-[N-(N’, N’-dimethyl aminoethane)-carbamoyl], cholesterol hydrochloride (DC-Chol), and zwitterionic lipid 1, 2-dioleoyl-snglycero-3-phosphoethanolamine (DOPE), in order to obtain cationic liposomes that were supposed to interact better with the negatively charged biofilm matrix. The liposomes were obtained by a thin film hydration method, with very high Tobramycin and peptide encapsulation efficiency, a positive surface charge (around 60 mV), and a size below 200 nm. Tobramycin-loaded liposomes significantly contributed to increasing the antibiofilm activity of Tobramycin, while peptide co-encapsulation did not. The results confirmed the efficacy of liposomes in delivering the antibiotic while enhancing the therapeutic efficacy and reducing the required effective dose of tobramycin. Nevertheless, the authors concluded that the role of the IDR-1018 peptide and other anti-biofilm peptides against lung infections should be further investigated [32].

The co-encapsulation of Tobramycin with 1,2-ethanedithiol bismuth (BiEDT) is another strategy that was pursued to increase antibiofilm activity and, ultimately, antibiotic efficacy. BiEDT antibacterial activity is not completely understood: it mainly relates to its interference with iron absorption, the reduction in alginate, lipopolysaccharides, and adhesion factor secretion by *P. aeruginosa*, which are constituents of the biofilm matrix together with the cytoplasmic accumulation of ethandithiol causing cell death [44]. The mechanism also affects quorum sensing in biofilm formation through N-Acyl homoserine lactones (AHLs or N-AHLs), a class of signalling molecules involved in bacterial quorum sensing [45]. Faeze Halwani and coll and Mahdiun and coll. experimented with two different nanosized drug delivery systems, namely liposomes and niosomes, respectively, and both the authors concluded that the co-encapsulation of Tobramycin with BiEDT significantly improved its microbicidal activity against *P. aeruginosa*. Halwani and coll. prepared liposomes with the thin film rehydration method. The liposomes were made from Disteroylphosphatidilcholine (DSPC) and cholesterol (2:1 molar ratio), supplemented with BiEDT in the lipid solution, and Tobramycin (being a hydrophilic compound) was added upon film hydration. The liposomes were neutral, with a size of around 900 nm. They demonstrated an ability to lower *Burkholderia cenocepacia* and *P. aeruginosa* MIC by three to eight-fold. Moreover, liposomes loaded with Tobramycin and 1,2-ethanedithiol bismuth demonstrated lower toxicity with superior antimicrobial activity to free tobramycin against *P. aeruginosa* and *B. cenocepacia* [30]. Mahdiun and coll. prepared niosomes made from cholesterol, Span 40, and Tween 40. They used the thin film hydration method, where BiEDT was dissolved in the lipid phase, following Tobramycin addition in the aqueous phase during lipid film rehydration. The niosomes were about 1 µm size making them stable and able to reduce Tobramycin’s minimal inhibitory concentration (MIC) four-fold [31]. Moreover, the niosomes loaded with Tobramycin and BiEDT inhibited AHL production by *P. aeruginosa*; thus, they affected the quorum sensing mechanism.

In summary, the available literature demonstrates that liposomes are an excellent delivery system for improving Tobramycin activity against Gram-negative microorganisms such as *P. aeruginosa* and their biofilm. The co-encapsulation of compounds that have mucolytic or antibacterial activity themselves can increase the antibacterial activity of Tobramycin. Controversial data are reported in the literature concerning liposome size correlation to antibacterial effects. Even if the positive liposome surface charge seems to be important for their penetration into negatively charged biofilms, the results reported in the literature highlight the fusegenity ability of liposomes due to their vesicular structure.

### 3.2. Polymer Nanoparticles

Polymer nanoparticles have been less investigated as carriers for Tobramycin. One of the reasons is that the polyesters most commonly used as nanoparticulate delivery systems, such as polylactide (PLA) and polylactide-co-glycolide (PLGA), are hydrophobic and their interaction with hydrophilic molecules similar to Tobramycin is poor, resulting in low encapsulation efficiency of the antibiotic drug. More efforts of scientists were paid to increase the drug loading into nanoparticles using different strategies. F. Ungaro and coll. investigated Tobramycin encapsulation into PLGA RG502H nanoparticles, which is more hydrophilic because the polymer chain end is not esterified. The authors achieved a suitable encapsulation efficiency of the antibiotic only when it was co-encapsulated with a hydrophilic polymer such as alginate or chitosan. Nevertheless, they obtained good results in delivering the Tobramycin-loaded nanoparticles to the lungs [46]. Another interesting experimental work was published by Hill and coll. where the authors compared Tobramycin encapsulation in PLGA RG502H and PLGA RG503 with the co-encapsulation of Tobramycin with dioctylsulfosuccinate (AOT) in the same polymers. Tobramycin encapsulation efficiency without AOT, was consistently higher for the RG502H nanoparticles compared to RG503, even if it was low. When Tobramycin was co-encapsulated with AOT, a very high encapsulation efficiency of the drug was achieved, around 90–96%, with both polymers. Moreover, the MIC of the encapsulated Tobramycin did not change with respect to free Tobramycin, but drug release was prolonged in time with prolonged antibacterial effects against *P. aeruginosa* [34].

The Nanoplex formation between Tobramycin alginate and chitosan was successfully investigated by Deacon and coll. In this case, two hydrophilic polymers were used; Tobramycin, due to its positive charge, interacted with the negatively charged alginate at pH 5 through van der Waals hydrogen bonds formation. The authors confirmed this interaction by isothermal titration calorimetric analysis. The Alginate/Tobramycin nanoplex was stabilized by chitosan, and the obtained nanoparticles were about 500 nm in size with a slight negative charge. Tobramycin antimicrobial activity was kept even after nanoencapsulation, and DNase functionalisation resulted in improved nanoparticle penetration into the mucus [35]. Moreover, the hydrophilic polymers were allowed to carry out nanoparticle preparation in very mild conditions, avoiding the use of organic solvents and achieving high drug payloads.

In summary, Tobramycin encapsulation into the polymer nanoparticles resulted in prolonged antimicrobial activity. However, encapsulation efficiency is a challenge and co-encapsulation with hydrophilic molecules or surfactants is useful.

### 3.3. Solid Lipid Nanoparticles (SLNs)

Solid lipid nanoparticles (SLN) are similar but not equal to liposomes. The terms liposomes and lipid nanoparticles are sometimes overlapped in the literature, but strictly speaking, this is not correct. They are both nanosized carriers made of lipidic material. However, SLN are solid particles at room and body temperature, while liposomes are vesicles made of lipid bilayers surrounding an aqueous core. Liposomes’ structure makes them quite unstable but is also their great advantage since the lipid bilayer gives liposomes fusogenic ability towards biological membranes. They should be lyophilized to overcome stability issues, obtained in a solid state, and rehydrated to be injected. SLN consists of solid lipids or a mixture of a solid lipid and a liquid lipid (nanostructure lipid carriers (NLC)) stabilized by the surfactants; they can have a micellar structure and usually have a single phospholipid outer layer that encapsulates the interior, which may be non-aqueous. As liposomes, SLN and NLC are biocompatible, and their manufacturing process is easy to scale up. In particular, NLC can be considered a second generation of the lipid nanoparticles. They are made from biodegradable lipids (solid and liquid) and emulsifiers whose blend results in the structural defects of solid lipids, lead to a less ordered crystalline arrangement. The peculiar solid state achieved permits to reach high drug payloads and prevent drug leakage [47,48]. NLC, loaded with Tobramycin, was prepared by Moreno and coll. as a pulmonary delivery system focusing on the treatment of cystic fibrosis patients. The NLC resulted in a very high encapsulation efficiency of about 93% and size always below 300 nm. They did not show cytotoxicity when tested on A549 and H441 as models of pulmonary epithelium cells. The authors tested Tobramycin-NLC permeation across the mucus using an artificial mucus with similar physico-chemical properties of CF patients’ mucus, and they concluded that Tobramycin-NLC was a stable carrier improving Tobramycin penetration into the mucus when combined with a mucolytic agent, such as Carabossimethylcysteine and, additionally, keeping the same MIC as the free drug [36]. A different use of Tobramycin SLN was proposed by Chetoni and coll. for the treatment of *P. aeruginosa* ocular infections. The authors prepared SLN through the micro emulsification of stearic acid, soya phosphatidylcholine, and soya tauroglycolate and added a 1:2 tobramycin:hexadecylphosphate complex. They obtained SLN with an 80 nm average size and negative zeta potential (−25 mV). The Tobramycin-SLN were able to halve free Tobramycin MIC, improving Tobramycin ocular bioavailability after their administration into rabbit eyes. The work was particularly interesting because (i) the micro emulsification preparation process resulted in very small nanoparticles (80 nm) with respect to other methods, (ii) Tobramycin is an antibiotic drug commonly used to treat ocular infections through instillation, (iii) the lowering of Tobramycin MIC was achieved with Tobramycin-SLN, (iv) SLN and was demonstrated to overcome the anatomical and physiological barrier of the eye [37,49]. 

In summary, SLN and NLC have been investigated as an alternative to liposomes for Tobramycin delivery. The authors achieved good results with the advantage of higher formulation stability, but biofilm and mucus penetration seemed to be better achieved with liposomes.

## 4. Other Carriers 

Hydrogels have been studied as carriers for Tobramycin, mostly in wound healing applications. In these treatments, small molecule drugs such as tobramycin may diffuse quickly on the surface of the infected wounds, with the potential risk of burst release and the development of drug resistance. To overcome these constraints, Tobramycin complexes have been studied. The most investigated polymers for this goal are alginate and chitosan, which are hydrophilic natural polymers with good moisture retention properties and well-known biocompatibility. Tobramycin is vehiculated in the hydrogels made from these polymers by complex formations with the polymer itself, such as in the case of alginate or by means of crosslink with oxidized dextran [50,51,52]. More in detail, Liang and coll. synthesized oxidized sodium alginate-tobramycin conjugate (OSA-TOB) with the tobramycin graft ratio in OSA-TOB of 13.8%. The OSA–TOB complex resulted in excellent healing rates, hemocompatibility, and cytocompatibility, also reducing the local inflammatory response and accelerated epithelium formation and collagen deposition. The hydrogel was tested against *S. epidermidis* (ATCC 12228), *S. aureus* (ATCC 6538), *P. aeruginosa* (ATCC 9027), and *E. coli* (ATCC 8739), which were selected as typical bacteria of infected wounds. The same principle of using Tobramycin was adopted for the crosslinker of alginate hydrogels and was applied by Xu and coll. whose goal was to obtain a hydrogel to be injected into deep wounds. Huang and coll. proposed self-healing hydrogels for burn wounds and healing applications that are susceptible to *P. aeruginosa* infections. The hydrogel was made from quaternized chitosan (QCS), oxidized dextran (OD), Tobramycin (TOB), and polydopamine-coated polypyrrole nanowires (PPY@PDA NWs), and it had good electrical conductivity and antioxidant activity. Moreover, the Schiff base crosslinks between the aminoglycoside antibiotic and OD enabled Tobramycin to be slowly released and the gel to be responsive to pH. The authors obtained excellent results in terms of prolonged antibacterial activity, wound healing, and biocompatibility of the Tobramycin-loaded hydrogel. In conclusion, all the proposed hydrogels seem to be potentially applicable as promising anti-infection wound dressings. However, they seem to be still quite preliminary works that need to be further deepened as long as formulation and application details are concerned, such as the sterility and osmotic pressure exerted by the hydrogel. These are unavoidable important parameters to be evaluated, above all, if the product is designed for injection.

Novel synthesized diblock copolymers were also investigated to complex Tobramycin as a model antibiotic drug, and the complex resulted in antimicrobial and antibiofilm activity. The copolymers consisted of a glycopolymer block containing either mannopyranose or galactopyranose pendant units, which was elongated with sodium 2-acrylamido-2-methyl-1-propanesulfonate (AMPS) to generate a polyanionic block, that enabled the complexation of cationic aminoglycoside antibiotic Tobramycin through electrostatic interactions [53]. This work is more focused on polymer synthesis than on the actual therapeutic application of the polymer–Tobramycin complex. However, it is a detailed example of polymer carrier design.

Polymer fibers loaded with Tobramycin were investigated and proposed for general biomedical purposes or, specifically, for tissue regeneration [54,55]. In the latter case, Tobramycin represents support to prevent and avoid infections after the implantation of a temporary polymeric scaffold whose main purpose is to promote tissue regrowth. In a more generic example, Tobramycin complexation with chitosan was exploited to produce submicron fibers by centrifugal spinning. Figure 4 reports the synthetic reaction that, similar to those of previous examples by Huang and coll., Liang and coll., and Xu and coll. exploit Schiff basis formations to achieve a Tobramycin complex sensitive to pH and able to release the antibiotic drug at pH lowering.

The authors added Polyethylene oxide (PEO) to the OCS–TOB complex in order to produce submicron fibres with different PEO contents by centrifugal spinning technique. The fibres were thoroughly characterized and demonstrated to be suitable for cell attachment and proliferation with antibacterial activity; mechanical properties and contact angle were tuneable depending on their composition and could be addressed depending on the final application [54].

The work of Rosalia and coll. starts from a medical need, i.e., peripheral artery occlusive disease, and exploits Tobramycin antibacterial activity to prevent infection due to the implantation in the human body of small-diameter vascular grafts made from biodegradable polymers. Polylactide-co-glycolide (PLGA) is here proposed as the polymer constituting the vascular graft, and attention is focused on mimicking those features of the extracellular matrix and radial artery in terms of fiber diameter, pore size, and matrix porosity. The tubular vascular grafts were manufactured by electrospinning a polymer solution in which the Tobramycin aqueous solution, supplemented with antioxidant sodium metabisulphite, was previously dispersed. This is because since PLGA is a hydrophobic polymer soluble in organic solvents, it does not solubilize the drug. The authors obtained excellent results in terms of drug encapsulation efficiency; the tubular grafts released an effective Tobramycin concentration and ensured a strong antibacterial activity over 5 days, making it a promising local drug delivery device.

Collagen was also proposed as a polymer for the delivery of Tobramycin to the cornea. The drug was chemically grafted to the surface of the collagen film using 1-ethyl-3-(3-dimethyl aminopropyl) carbodiimide (EDC) and N-hydroxysuccinimide (NHS) as cross-linking agents. The mechanical properties, light transmittance, antibacterial property, and biological properties of the cross-linked film were characterized. The results show that the film has similar water absorption, mechanical properties, and light transmittance to the native cornea, with excellent antibacterial effects and biocompatibility. After implantation in the rabbit eye, the corneal rejection reaction, neovascularization, and keratoconus were not observed within 3 months. This cross-linked film shows the potential to solve the bacterial infection possibly rising after keratoplasty [56].

In summary, hydrogels, films, and fibers have been studied mainly to achieve the prolonged release of tobramycin in localized administration, such as in wound healing, or to treat infections in specific organs, such as the eyes. The effort of researchers was to deliver Tobramycin complexed with polymers, and pH-sensitive complexes seem to be promising. The works published in the literature seem to be preliminary, and the field is open to the research of new polymers that could optimize Tobramycin delivery.

## 5. Discussion

The delivery of Tobramycin through the nanosized drug delivery system was studied with the main goal of overcoming antimicrobial resistance, which is a challenge nowadays. The strategy involves encapsulating Tobramycin in order to better permeate across the bacterial membranes. This result is particularly interesting for Gram-negative bacteria that possess an outer membrane and an inner membrane surrounded by a peptidoglycan layer. The membrane structure and the presence of efflux pumps and highly selective porins protect the Gram-negative bacteria, making drug intracellular targeting more difficult, especially for hydrophilic drug molecules. For these reasons, lipophilic carriers and, above all, amphiphilic ones, such as liposomes, were preferentially studied in the past few years with good results (see Table 1). Nevertheless, SLN and NLC seem to be an improved alternative to liposomes because they are more stable with the same ability to permeate across bacteria membranes. Moreover, NLCs are characterized by their ability to load high amounts of drugs. Additionally, polymer nanoparticles could be an alternative to liposomes, but in this case, a combination of hydrophobic and hydrophilic polymers seems to achieve the best results. The hydrophobic polymer, such as PLGA, stabilizes the NP and promotes its permeation across the bacterial membranes, while the hydrophilic polymer, i.e., Alginate, helps in increasing drug content.

Moreover, the nanosized carriers are permitted to successfully combine Tobramycin with other drugs and/or compounds in order to enhance its penetration across the bacteria biofilm. The most studied therapeutic need investigated was cystic fibrosis, a chronic pathology that frequently infects with *P. aeruginosa*. The bacterium produces a thick biofilm made from cell clusters and exopolysaccharides that interact with anionic host polymers in sputum, making a thick and resistant barrier to drugs [57]. Thus, the most effective nanosized carriers’ administration route proposed is pulmonary via the trachea; the formulation can be a suspension or powder aerosol. Another application of nanosized carriers loaded with Tobramycin is to treat eye infections. Even in this case, the goal is to achieve effective therapy in a site that is difficult to be reached and where it is difficult to maintain suitable drug therapeutic levels. The administration route proposed is topical, directly into the eye, and the nanosized carriers, due to their mucoadhesive properties (i.e., polymer NPs), or their penetration enhancement (i.e., SLN), are able to achieve this goal. Moreover, tobramycin-loaded films, hydrogels, or nanofibers can be suitably applied in eye infection therapies by corneal insertion.

Another strategy proposed in the literature resides in synthesizing antibiotic hybrids. These are defined as synthetic constructs of two or more pharmacophores belonging to an established agent known to elicit a desired antimicrobial effect. An interesting and exhaustive review has been recently published on this topic, where examples of antibiotic hybrid drugs and prodrugs were reported. However, hybrid strategies are challenging, both from the standpoint of the pharmacokinetics of the antibiotic hybrid and of hybrid synthesis, even because therapeutic agents typically possess dissimilar molecular stabilities and reactivities under different preparative conditions [58]. Thus, exploiting drug delivery for improving Tobramycin, and in general antibiotic activity, and overcoming antibiotic resistance seems an easier way to achieve good results.

## 6. Conclusions

Tobramycin is a conventional aminoglycoside antibiotic that has been thoroughly studied to be repurposed in drug delivery systems. The case of Tobramycin is particularly interesting because nanosized carriers, when loaded with this drug, can greatly improve the efficacy of topical therapies, such in cystic fibrosis or ocular infections.

## 7. Patents

The authors did not file any patent resulting from the work reported in this manuscript.

## Figures and Tables

**Figure 1 ijms-23-14080-f001:**
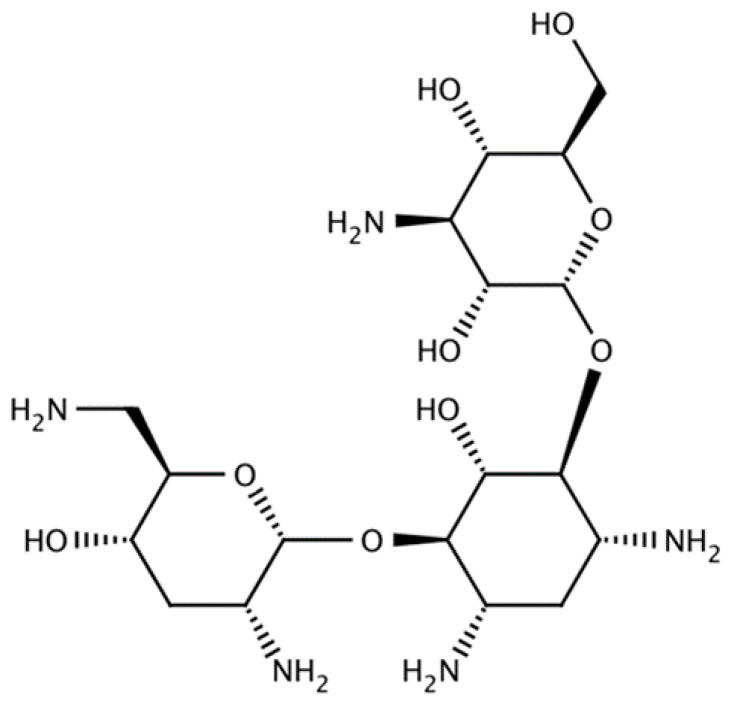
Tobramycin molecular structure.

**Figure 2 ijms-23-14080-f002:**
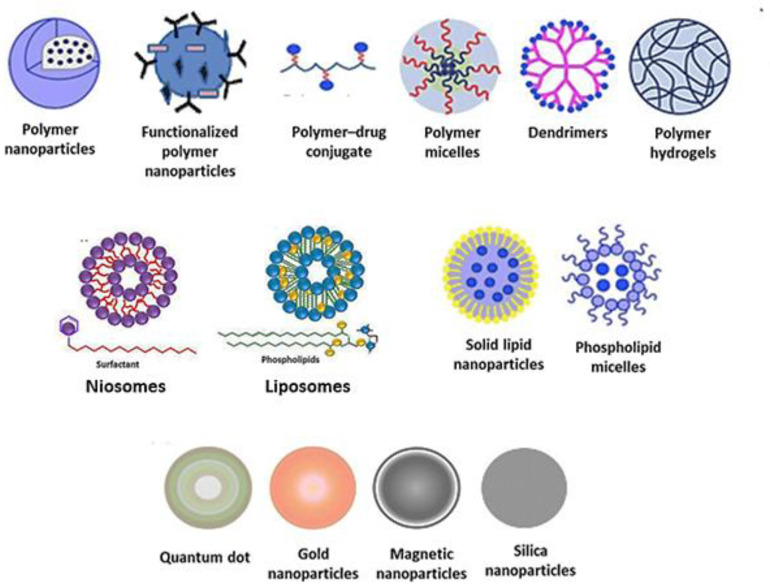
Nanosized carriers.

**Figure 3 ijms-23-14080-f003:**
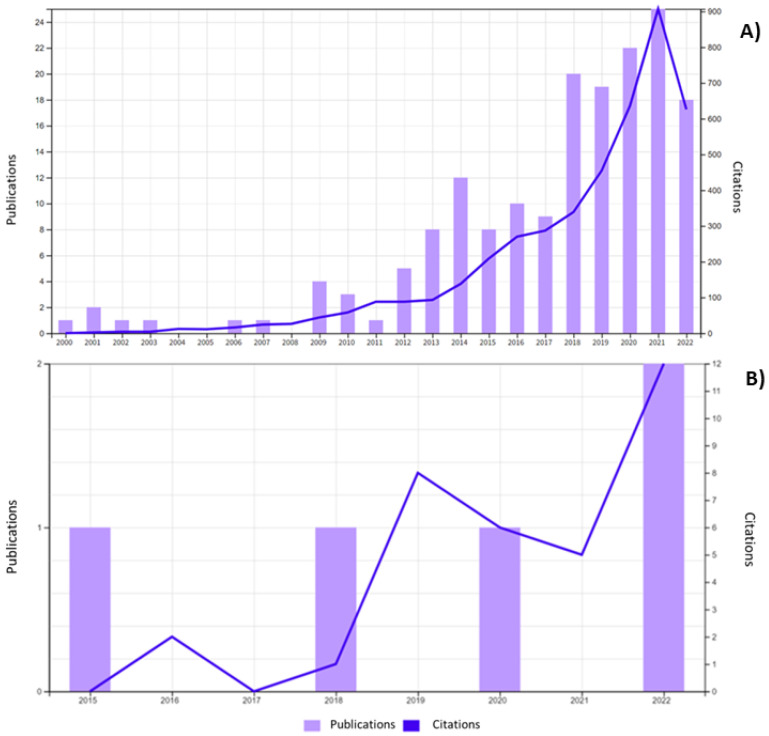
Outcomes of a Web of Sciences search from 1 January 2000 to September 2022, accessed on 4 October 2022; (**A**) By the term “Tobramycin nanoparticles”; (**B**) By the term “Tobramycin nanofibers”.

**Figure 4 ijms-23-14080-f004:**
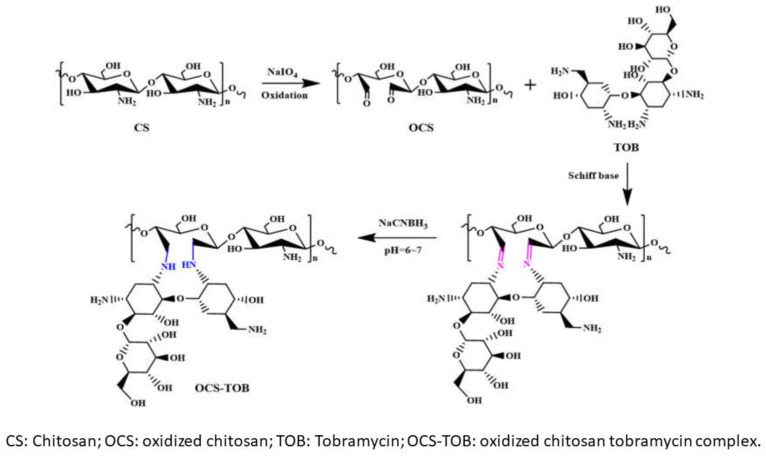
Synthetic pathway of oxidized chitosan (OCS) and oxidized chitosan–Tobramycin complex (OCS-TOB) [54].

**Table 1 ijms-23-14080-t001:** Examples of Tobramycin loaded nanocarriers, their composition and efficacy.

Nanocarrier Type /Admin. Route	Composition and Encapsulation Efficiency (EE)	Encapsulated Tobramycin Performance (Compared to Free Drug)	Refs.
Liposomes/aerosol	Fluidosomes^TM^	CFU lowering of about 300 time against isolate of *P. aeruginosa,* PA 508.	[25]
Liposomes/aerosol	10:1 molar ratio dipalmitoylphosphatidylcholine (DPPC): dimyristoylphosphatidylglycerol (DMPG)	Significant increase in *t*1/2_ from the lungs against BC. **	[28]
Liposomes/intratracheal	10:1 molar ratio DPPC:DMPG	Significant increase in *t*1/2_ from the lungs against isolate of PA **,* PA 508.	[29]
Liposomes	2:1 molar ratio Disteroylphospahtidylcholine (DSPC): cholesterol, bismuth-ethanedithiol	MIC reduction by 3 to 8-fold against PA and BC, respectively.	[30]
Niosomes	Span 40, Tween 40, cholesterol, ratio 70:30	A 4-fold MIC reduction against clinical strains of PA ATCC 27583. A total of 80% inhibition of biofilm formation. Inhibition of N-Acyl homoserine lactones (AHL) production.	[31]
Liposomes	3_-[N-(N’, N’-dimethyl aminoethane)-carbamoyl], cholesterol hydrochloride (DC-Chol), 1, 2-dioleoyl-snglycero-3-phosphoethanolamine (DOPE). Anti-biofilm peptide (IDR 1018)	Significant increase in antibiofilm activity against PA and its biofilm, except for IDR1018 addition.	[32]
Liposomes	1,2-Dimyristoyl-sn-glycero-3-phosphoethanolamine (DMPE), DPPC, cholesterol. N-acetylcysteine	Significant increase in antimicrobial and antibiofilm activity gainst *E. coli*, *K. pneumoniae*, and *A. baumannii.*	[33]
Polymer nanoparticles / in vitro	Polylactide-co-glycolide (PLGA) Resomer RG503, PLGA RG 502H, dioctylsulfosuccinate (AOT). 95% encaps efficiency (EE).	MIC kept same values of not encapsulated drug against PA.	[34]
Polymer nanoparticles/ in vitro	Sodium alginate, chitosan (low MW). Dornase alfa (DNase). 45% EE	Slight significant improvement of DNase tobramycin NPs antimicrobial and antibiofilm activities against PA.	[35]
SLN/ in vitro and intratrachea to mice	Glycerol distearate, type I, glyceryl dibehenate, Poloxamer 188. 93–99% EE	Sustained drug release into lungs.	[36]
SLN/intraocular to rabbits	Stearic acid, soya phosphatidylcholine, soya tauroglycolate. 95% EE. 2.5% drug loading	A total 3-fold bioavailability increase.	[37]

* PA: P. *aeruginosa* ** *Burkholderia*
*cenocepacia*.

## Data Availability

Not applicable.

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
