# Peer review of "Tobramycin Nanoantibiotics and Their Advantages: A Minireview"

_ijms, 2022, doi:10.3390/ijms232214080_

Round 1

Reviewer 1 Report

Dear Authors, your manuscript titled "Tobramycin nanoantibiotics and their advantages: a minireview" is complete and provides an interesting overview of the potential of using scaffolds for tobramycin delivery.

Do you think that these different kinds of drug delivery could be used for other aminoglycosides too?

line 142 please re-read the sentence because it seems incomplete (in the brush border of the.)

line 227 please correct "cepacian" with "cepacia"

lines 507 and 526 I suggest to put "topical or local" instead of  localized

Author Response

IJMS Paper: Tobramycin nanoantibiotics and their advantages: a minireview

Reviewer 1

Dear Authors, your manuscript titled "Tobramycin nanoantibiotics and their advantages: a minireview" is complete and provides an interesting overview of the potential of using scaffolds for tobramycin delivery.

Do you think that these different kinds of drug delivery could be used for other aminoglycosides too?

ANSWER: We thank Reviewer 1 for his/her positive feedback. Aminoglycoside antibiotics share the same molecular structure, composed by an aminocyclitol core linked to several aminated sugars. Due to this common structure, the physical and chemical properties of different aminoglycosides are similar, suggesting the possibility to apply nanosized drug delivery systems also to other types on antibiotics. Moreover, antibiotic resistance and severe systemic side effects represent a concern not only for the therapeutic use of tobramycin, but also for other aminoglycoside, among others gentamycin and streptomycin. It is therefore desirable the design of new nanosized delivery systems suitable for other aminoglycoside antibiotics. On this behalf, we recommend the exhaustive overview on nanostructured aminoglycosides by S. Bera & D. Mondal (2022) (DOI:10.1021/acsomega.1c04399).

line 142 please re-read the sentence because it seems incomplete (in the brush border of the.)

line 227 please correct "cepacian" with "cepacia"

lines 507 and 526 I suggest to put "topical or local" instead of  localized  

ANSWER: the required corrections at lines 142, 227, 507 have been done in the manuscript revised version.

Reviewer 2 Report

The paper is well written and it is easy to follow...

Author Response

Dear reviwer,

our point by point answers are attached here below.

Reviewer 2

The paper is well written and it is easy to follow...

 ANSWER: We thank Reviewer 2 for his/her positive feedback and for the in-text suggestions. We incorporated most of the corrections, but decided also to maintain unchanged some sentences, to avoid interfering with the fluency and clarity of the text.  

Reviewer 3 Report

The authors have worked very nicely to write a review on a very informative topic. The article is well written. The article needs some minor corrections, before proceeding for the possible acceptance.

My comments are:

Line 12: write abbreviation (AMR) after antimicrobial resistance.

Line 13: write “Gram-negative” instead of Gram-

Line 18: write “and” before Citrobacter

Line 24: write abbreviation (MDR) after multidrug resistant.

Line 34 and 35: No need to put here full form of AMR and MDR.

Line 68: No need to put here full form of NPs

Line 117: Italicize “Streptomyces tenebrarius”

Table 1: Cite table 1 in the text above the table.

References need to be corrected. For example, reference 1.

The authors can cite more references to strengthen the statements. For example, in Introduction paragraph 1, authors can cite for literature mentioned before line 35.

Best wishes!

Author Response

Dear reviewer,

the point by point answers to your comments are attached here below.

The authors have worked very nicely to write a review on a very informative topic. The article is well written. The article needs some minor corrections, before proceeding for the possible acceptance.

My comments are:

Line 12: write abbreviation (AMR) after antimicrobial resistance.  

Line 13: write “Gram-negative” instead of Gram-     

Line 18: write “and” before Citrobacter     

Line 24: write abbreviation (MDR) after multidrug resistant.  

Line 34 and 35: No need to put here full form of AMR and MDR.

Line 68: No need to put here full form of NPs  

Line 117: Italicize “Streptomyces tenebrarius”  

Table 1: Cite table 1 in the text above the table.

ANSWER: We thank Reviewer 3 for his/her positive feedback. We have welcomed the comments and suggestions and we modified the text at lines 12, 13, 18, 24, 68, 117 and Table 1 accordingly. We decided not to modify the text at lines 34 and 35 for clarity purposes.

References need to be corrected. For example, reference 1.

ANSWER: reference 1 has been corrected. We hope now is allright, since reference 1 refers to a publication of World Health Organization, European Centre for Disease Prevention and Control  and does not have names for single authors.

The authors can cite more references to strengthen the statements. For example, in Introduction paragraph 1, authors can cite for literature mentioned before line 35.  

ANSWER: we think that reference 1, “World Health Organization, European Centre for Disease Prevention and Control  , Antimicrobial resistance surveillance in Europe : 2022 : 2020 data. Publications Office of the European Union: 2022. ISBN 978-92-9498-552-1”,  gives a complete updated view of bacterial infections and  antibiotic resistance together with its monitoring. Another interesting paper is “Ramanan Laxminarayan, Anil Deolalikar, Keshav Desiraju, Didier Pittet, Stephen Tollman, Mary E. Wilson, The State of the World's Antibiotics 2021, Chapter 1. Changing Patterns in Antimicrobial Resistance, edited by Sally Atwater,The Center for Disease Dynamics, Economics & Policy (CDDEP), Inc. 2021. https://creativecommons.org/licenses/by-ncnd/4.0/